# Using C2X to Explore the Uncertainty of In Situ Chlorophyll-a and Improve the Accuracy of Inversion Models

Wen Li [1,2], Yadong Zhou [1], Fan Yang [1,2], Hui Liu [1], Xiaoqin Yang [3], Congju Fu [1,2] and Baoyin He [1,*]

1. Key Laboratory for Environment and Disaster Monitoring and Evaluation of Hubei Provincial, Innovation Academy for Precision Measurement Science and Technology, Chinese Academy of Sciences, Wuhan 430077, China; liwen21@mails.ucas.ac.cn (W.L.); zhouyadong@apm.ac.cn (Y.Z.); yangfan@apm.ac.cn (F.Y.); liuhui@apm.ac.cn (H.L.); fucongju@apm.ac.cn (C.F.)
2. University of Chinese Academy of Sciences, Beijing 100049, China
3. Hydrological and Water Resources Survey Bureau of Wuhan, Wuhan 430071, China; xiaoqin@whpyinfo.com
* Correspondence: heby@apm.ac.cn

**Abstract:** Quality water plays a huge role in human life. Chlorophyll-a (Chl-a) in water bodies is a direct reflection of the population size of the primary productivity of various phytoplankton species in the water body and can provide critical information on the health of water ecosystems and the pollution status of water quality. Case 2 Regional CoastColour (C2RCC) is a networked atmospheric correction processor introduced by the Sentinel Application Platform for various remote sensing products. Among them, the Extreme Case-2 Waters (C2X) process has demonstrated advantages in inland complex waters, enabling the generation of band data, conc_chl product for Chl-a, and kd_z90max product for Secchi Depth (SD). Accurate in situ data are essential for the development of reliable Chl-a models, while in situ data measurement is limited by many factors. To explore and improve the uncertainties involved, we combined the C2X method with Sentinel-2 imagery and water quality data, taking lakes in Wuhan from 2018 to 2021 as a case. A Chl-a model was developed and validated using an empirical SD model and a neural network incorporating Trophic Level Index (TLI) to derive the predicted correction result, Chl-a_t. The results indicated that (1) the conc_chl product measured by C2X and in situ Chl-a exhibited consistent overall trends, with the highest correlation observed in the range of 2–10 μg/L. (2) The corrected Chl-a_t using the conc_chl product had a mean absolute error of approximately 10–15 μg/L and a root-mean-square error of approximately 8–10 μg/L, while using in situ Chl-a had a root-mean-square error (RMSE) of approximately 15 μg/L and a mean absolute error (MAE) of approximately 20 μg/L; both errors decreased by double after correction. (3) The correlation coefficient (R) between Chl-a_t and each data point in the Chl-a model results was lower than that of SD-a_t with each data point in the SD model results. Additionally, the difference in R-value between Chl-a_t and each data point (0.45–0.60) was larger than that of SD-a_t with each data point (0.35–0.5). (4) When using corrected Chl-a_t data to calculate the TLI estimation model, both RMSE and MAE decreased, which were 1μg/L lower than those derived from uncorrected data, while R increased, indicating an improvement in accuracy and reliability. These findings demonstrated the presence of in situ errors in Chl-a measurements, which must be acknowledged during research. This study holds practical significance as some of these errors can be effectively corrected through the use of C2X atmospheric correction on spectral bands.

**Keywords:** remote sensing; Chlorophyll-a; Case2eXtreme; lakes of Wuhan

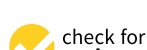



## 1. Introduction

Water is the fundamental source of life, and the water resources available to humans primarily comprise rivers, lakes, and groundwater. Among these freshwater sources, lakes and reservoirs are particularly crucial for human survival and well-being; their exploitation can significantly impact both human productivity and other organisms' livelihoods [1]. As

a crucial component of inland aquatic ecosystems, closed inland lakes are highly susceptible to anthropogenic climate change. The water quality of these lakes is increasingly imperiled by human activities and the impacts of climate change [2]. Therefore, it is imperative to monitor water quality and implement timely measures for its improvement to ensure the sustainable utilization of water resources.

Chlorophyll-a (Chl-a), a pigment commonly found in green plants and algae, is utilized by these organisms to absorb sunlight and convert it into chemical energy. Chl-a is a crucial indicator for evaluating algal and plant biomass, water quality, and ecosystem health in aquatic environments. Elevated levels of Chl-a typically signify an abundance of nutrients in the water body, making it a key parameter for monitoring nutrient status and overall water quality [3–5]. Chl-a provides critical information to help develop effective conservation and management strategies to ensure the health of water ecosystems. Thiemann et al. [6] used Indian satellite remote sensing data (IRS-1C) and measured spectral data to invert the Chl-a content of German lakes and finally assess the trophic status of the lakes. Rahim et al. [7] used Chl-a to predict water quality changes in Lake Mikri Prespa in Greece. Yang et al. and Peterson [8,9] have provided a scientific basis for diagnosing eutrophication in inland lakes by designing Chl-a retrieval models. First, errors may lead to bias in the accurate assessment of water quality, which in turn affects the judgment of water body safety and treatment decisions. Second, the error may mask the potential pollution problems in the water body, resulting in a lack of accurate understanding of the health of the water environment and its impact on the ecosystem. In addition, errors may also affect the comparability and continuity of water quality monitoring data, making the analysis and prediction of long-term trends in water bodies limited. Most importantly, errors can lead to incorrect judgments and treatments that pose potential risks to human health and ecosystems. Therefore, accurate measurement of water quality parameters is essential to protect the water environment and human health.

Through the Chl-a retrieval model, a more comprehensive analysis of the water quality status can be obtained. Accurate in situ water quality parameters serve as the foundation for experiments and model building. However, obtaining measurement data is constrained by multiple factors such as climate, environment, and methods which can easily result in measurement errors and significant uncertainty. Uncertainty in in situ Chl-a is significant due to various factors, including the storage time of water samples, wind speed, sampling depth, and the level of operational discipline of samplers. The extensive workload associated with in situ sampling creates temporal inconsistencies between field measurements and satellite overpasses. The concentrations of water quality components exhibit a vertical stratification effect. For example, algae fluctuate up and down at different times of the day, depending on water temperature and their own growth characteristics, resulting in large differences in Chl-a over short periods, leading to a mismatch between spectral information and in situ data [10]. On short timescales, weather factors cause drastic spatial changes in the water quality components; on long-time scales, each sampled environment is different and may also be affected by climate change, increasing the uncertainty of the measured data accumulated over time [11,12]. All the above factors can cause poor synchronization between in situ data and spectral data, resulting in fluctuations in water quality parameter concentration. Therefore, eliminating parameter uncertainty is crucial for improving the accuracy of Chl-a models.

Remote sensing technology boasts the advantages of affordability, rapidity, synchronicity, and wide coverage, making it a popular tool for monitoring lake chlorophyll. Among various satellite data options, Sentinel-2 stands out with its high temporal and spectral resolution that enables accurate assessment of lake water quality [13–15]. A critical step in processing Sentinel-2 data is atmospheric correction, which converts radiometric brightness to actual surface reflectance to eliminate errors caused by atmospheric scattering, absorption, and reflection. The Case 2 Regional CoastColour (C2RCC) provided by the ESA's Sentinel Application Platform (SNAP) serves as a valuable tool for atmospheric correction. Its neural network inversions are trained on simulated atmospheric reflectance

data, integrating optical properties from around the world. C2RCC employs radioactive transfer theory to construct accurate correction [16]. Hydrolight and vector Successive Order of Scattering (SOS) atmospheric models are involved, incorporating aerosol optical properties derived from NASA AERONET-OC measurements. The processor is categorized into Case 2 (C2), Case2eXtreme (C2X), and Case2eXtreme-complex (C2X-complex) to cater to different multispectral satellites and regions. Among these methods, the C2X processor is found to be more suitable for turbid and optically complex inland lakes [17]. The final products obtained consist of a variety of data covering remote sensing reflectance at different wavelengths, kd_z90max product, conc_chl product, and suspended matter concentration. The reliability of water quality parameters obtained by C2RCC has been demonstrated in previous studies [18–20]. Moreover, C2RCC exhibits excellent performance in terms of Chl-a and Secchi depth (SD) [21–23]. Carsten et al. [16] confirmed the effectiveness of C2RCC in some water bodies. Jesús Soriano-González et al. [17] explored the performance of C2RCC in inland waters of the Eastern Iberian Peninsula using Sentinel-2 multispectral images, which showed that C2RCC had an excellent application effect in water quality monitoring. The C2RCC has the potential to revolutionize the field of water quality remote sensing by offering exceptionally accurate and precise data regarding water quality parameters. Therefore, the utilization of band data acquired through C2X atmospheric correction is pivotal in the modeling correction of Chl-a in this study.

Water quality parameter inversion methods are generally categorized into semi-analytical, empirical/semi-empirical, and machine learning algorithms. The main methods used in this paper are empirical methods and machine learning. The empirical method is based on the statistical correlation between the in situ water quality data and remote sensing data to derive the water quality inversion model, select the optimal waveband or combination of waveband data and build the inversion model [24]. It is widely favored among scholars due to its speed and ease of use. Song et al. [25] used both empirical regressions and neural networks to analyze the relationship between the concentrations of water parameters and the satellite radiance signals. Escoto et al. [26] developed an empirical least squares regression model to estimate monitoring of the Pasig River located in the Philippines and used k-fold cross-validation to obtain good model accuracy results. Machine learning is also a frequently used model in water quality remote sensing research, and its application in the field of water quality remote sensing is developing rapidly due to its powerful data mining capability [27], which can use a variety of artificial intelligence algorithms to simulate the water body radiation transmission process. The advantage of this approach is the ability to quickly obtain predicted values for water quality parameters after model training. However, a significant amount of real measurement data is required for effective model training, and the accuracy of the model heavily relies on the quality of the training sample. The results of Sun et al. [28–30] suggested that proximal remote sensing combined with machine learning algorithms has great potential for monitoring water quality in inland waters. He et al. [31] and Elsayed et al. [32] further demonstrated that artificial neural networks are an effective tool for eutrophication assessment in reservoirs.

The band data obtained through C2X atmospheric correction serve as the fundamental basis for all aspects of this study. This study aimed to investigate the uncertainty associated with in situ Chl-a and improve it to increase its accuracy. The validated conc_chl data and in situ Chl-a data were compared to develop an empirical regression equation-based model for Chl-a. To aid validation, the TLI neural network modeling was used with the empirical regression SD model, taking advantage of the small measurement error in in situ SD and the more comprehensive nature of TLI. Finally, accuracy between various datasets was analyzed to clarify uncertainty in in situ Chl-a.

## 2. Study Area and Data

### 2.1. Study Area

Wuhan, situated at the confluence of the Yangtze River and its largest tributary, the Han River (113°41′~115°05′ E, 29°58′~31°22′ N), serves as the capital city of Hubei

province. The region features a north subtropical monsoon climate with hot and rainy seasons; precipitation primarily occurs between June and August. Wuhan boasts China's most extensive collection of lakes and is renowned as "The City of Hundred Lakes". There are a total of 166 lakes that serve primarily for irrigation, fishery farming, tourism, and industrial and urban domestic water purposes (Figure 1). In recent years, due to both climate change and human activities, eutrophication has become an increasingly pressing issue for Wuhan's lakes [33–35].

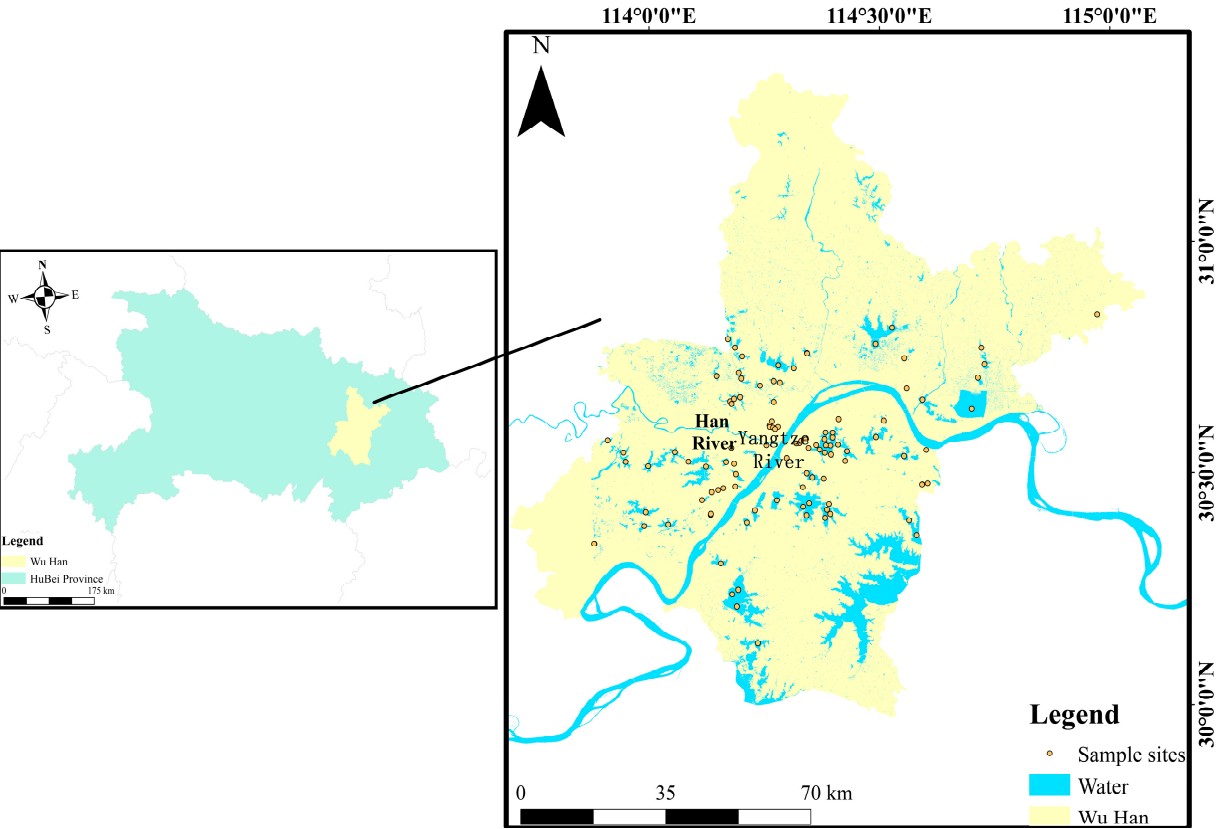

**Figure 1.** Location and distribution of sampling sites within the study area.

*2.2. Remote Sensing Data Acquisition and Pre-Processing*

Sentinel-2, developed by the European Space Agency (ESA), is a high-resolution multispectral imaging satellite for monitoring changes in surface conditions and provides data support for the Global Environmental and Security Information Service (GESIS) [15,36,37]. It consists of two satellites, S2A, and S2B, staggered by 180° in the same sun-synchronous orbit. Each satellite is equipped with an onboard multispectral instrument (S2-MSI) capable of covering 13 operating bands, including 4 visible bands, 6 near-infrared bands, and 3 short-wave infrared bands [36]. The revisit cycle for a single satellite is 10 days, but with two satellites complementing each other, the revisit period is reduced to 5 days. By acquiring images at different spatial resolutions (10 m, 20 m, and 60 m), Sentinel-2 can reflect the detailed features of the surface. In addition, Sentinel-2 possesses three bands (centered at 664 nm, 705 nm, and 745 nm) that are very effective at capturing Chl-a information from water. Therefore, it stands out as one of the most suitable satellite data sources for monitoring water quality in small and medium-sized bodies of water (Table 1).

**Table 1.** Sentinel-2 band information.

| Sentinel-2 Bands | Sentinel-2A | | Sentinel-2B | | |
| --- | --- | --- | --- | --- | --- |
| | Central Wavelength (nm) | Bandwidth (nm) | Central Wavelength (nm) | Bandwidth (nm) | Spatial Resolution (m) |
| Band 1—Coastal aerosol | 442.7 | 21 | 442.2 | 21 | 60 |
| Band 2—Blue | 492.4 | 66 | 492.1 | 66 | 10 |
| Band 3—Green | 559.8 | 36 | 559.0 | 36 | 10 |
| Band 4—Red | 664.6 | 31 | 664.9 | 31 | 10 |
| Band 5—Vegetation red edge | 704.1 | 15 | 703.8 | 16 | 20 |
| Band 6—Vegetation red edge | 740.5 | 15 | 739.1 | 15 | 20 |
| Band 7—Vegetation red edge | 782.8 | 20 | 779.7 | 20 | 20 |
| Band 8—NIR | 832.8 | 106 | 832.9 | 106 | 10 |
| Band 8A—Narrow NIR | 864.7 | 21 | 864.0 | 22 | 20 |
| Band 9—Water vapor | 945.1 | 20 | 943.2 | 21 | 60 |
| Band 10—SWIR–Cirrus | 1373.5 | 31 | 1376.9 | 30 | 60 |
| Band 11—SWIR | 1613.7 | 91 | 1610.4 | 94 | 20 |
| Band 12—SWIR | 2202.4 | 175 | 2185.7 | 185 | 20 |

To ensure the acquisition of high-quality data, images with low cloud coverage need to be selected. Additionally, it is imperative to select images with minimal cloud coverage. In addition, efforts were made to choose images that closely align with the dates of measured data and maintain a maximum interval of three days between image date and data acquisition for optimal accuracy in analysis.

The Sentinel-2 data were obtained from the Copernicus data-sharing service (https://scihub.copernicus.eu/ (accessed on 1 January 2022) and comprised both L1C and L2A level images. The L2A level images are processed with Sen2Cor atmospheric correction, which served as a reference for C2X atmospheric correction so that it was directly calculated for application to the control group in the TLI model in this study.

The L1C level data, without atmosphere correction, is the primary data for this study which require C2X processing. In the SNAP, the WGS84-49N projection coordinate system was selected, and the image was resampled to 10m resolution. For lakes in Wuhan, the salinity could be set to 0.18. Thus, C2X-processed remote sensing products were obtained for further research and analysis to support subsequent research.

### 2.3. Measured Water Quality Component Data on the Ground

The ground sample points mainly recorded the latitude and longitude information and water quality parameter (WQP) data. The in situ WQPs contain Chl-a, SD, total phosphorus (TP), total nitrogen (TN), and chemical oxygen demand using manganese dioxide as the oxidizing agent ($COD_{Mn}$). To reflect the spatial distribution of water quality better, we utilized ArcGIS 10.8 and other software to obtain vector information points. From there, we extracted the atmospherically corrected Sentinel-2 data information and attached corresponding sample point water quality parameter data. After calculation, analysis, and the elimination of abnormal values and other links, the construction of the sample database was completed. The final dataset covers mainly from November 2018 to January 2021, with a total of 210 recorded samples (Table 2). The sample bank data were randomly divided into two subsets: a training sample group (tra) and a testing sample group (tes) with a ratio of 7:3.

#### 2.3.1. Method of Obtaining Chl-a

The spectroFiguremetesc method of Chl-a determination is the standard in the environmental monitoring sector in China. The Chl-a was measured in lakes of Wuhan according to *the National Environmental Protection Standard of the People's Republic of China HJ 897-2017—SpectroFiguremetesc method for the determination of chlorophyll-a in water quality* [38]. The experimental procedure involved filtering a specific quantity of the sample through a membrane to retain algae, grinding and crushing algal cells, extracting chlorophyll with

an acetone solution, centrifuging and separating it, measuring the extract's absorbance at 750 nm, 664 nm, 647 nm, and 630 nm, respectively, and calculating it according to the prescribed formula (μg/L).

**Table 2.** Sampling point acquisition and corresponding image transit dates.

| Satellite | Sampling | | | | |
|---|---|---|---|---|---|
| Transit Date | Year | Month | Date | Number of Samples | Total Number of Samples |
| | 2018 | 10 | 31 | 2 | |
| 1 November 2018 | 2018 | 11 | 1 | 29 | 53 |
| | 2018 | 11 | 2 | 19 | |
| | 2018 | 11 | 3 | 3 | |
| | 2019 | 5 | 5 | 13 | |
| 8 May 2019 | 2019 | 5 | 6 | 6 | 20 |
| | 2019 | 5 | 9 | 1 | |
| 5 March 2020 | 2020 | 3 | 4 | 3 | 7 |
| | 2020 | 3 | 6 | 4 | |
| | 2020 | 8 | 31 | 2 | |
| 1 September 2020 | 2020 | 9 | 1 | 44 | 69 |
| | 2020 | 9 | 2 | 21 | |
| | 2020 | 9 | 3 | 2 | |
| | 2020 | 11 | 2 | 3 | |
| 3 November 2020 | 2020 | 11 | 3 | 4 | 12 |
| | 2020 | 11 | 4 | 5 | |
| | 2021 | 1 | 4 | 15 | |
| 4 January 2021 | 2021 | 1 | 5 | 16 | 49 |
| | 2021 | 1 | 6 | 16 | |
| | 2021 | 1 | 7 | 2 | |

The measurement method's control should be noted to only allow for the detection of Chl-a levels above 2 μg/L. In addition, the water quality of Wuhan's lakes has resulted in the significant maintenance of Chl-a at elevated levels [39–42]. The Chl-a in South Lake and East Lake were generally above 10 μg/L. A study by Yang et al. [35] in East Lake from 1987 to 2018 showed that Chl-a generally did not fall below 7.81 μg/L. In addition, the C2X method imposes a constraint on the range of Chl-a values (0–100 μg/L), which implies that the maximum cannot exceed 100 μg/L when using Chl-a [43]. Therefore, combining the characteristics of the study area with the study method, samples below 7 μg/L and above 100 μg/L were excluded from our study to reduce errors in the Chl-a values.

The final sample pool consisted of 210 carefully selected data points after eliminating any Chl-a values that were deemed inappropriate.

Conc_chl is the product representing Chl-a in μg/L, calculated from the intrinsic optical property iop_apig. Iop_apig is the absorption coefficient of the phytoplankton pigment at 443 nm in $m^{-1}$.

According to the SNAP user manual, the formula for calculating the conc_chl product is as follows:

$$conc\_chl = iop\_apig^{1.04} \times 21 \tag{1}$$

2.3.2. Method of Obtaining Secchi Depth

The Secchi Depth (SD) is the degree to which natural light can penetrate and reflect off the waters of a lake, providing insight into its clarity and turbidity. SD serves as an indicator of the presence of organic and inorganic matter in the water, making it a valuable tool for researchers studying parameters such as Chl-a that are critical to assessing lake water quality [44–46].

It was measured using a black and white Secchi disc, which is suspended in the water, and as it slowly sinks in the water until the black and white lines of the Secchi disc are not visible, the length of the suspended line is observed, while the wet part is recorded (m).

Kd_z90max is extracted from satellite images, representing 90% of outgoing water irradiance from the depth of the water body column (m) and can be regarded as a product of SD.

### 2.3.3. Trophic Level Index

In this study, the Trophic Level Index (TLI) method of the China Ecological Environment Bureau was employed [7]. The trophic level of a water body is assessed based on a set of relevant indicators and their interrelationships, which are obtained by calculating the values of five parameters: Chl-a, SD, TN, TP, and $COD_{Mn}$. The calculation formula in the Technical Provisions, issued by the General Station of Environmental Monitoring of the Ministry of Environmental Protection. The expressions are as follows:

$$TLI(\Sigma) = \Sigma Wj \cdot TLI(j) \tag{2}$$

where *Wj* is the trophic state index for the *j*-th parameter, and *TLI(j)* is the trophic state index for the *j*-th parameter. Using Chl-a as the base parameter, the normalized correlation weight for the *j*-th parameter is calculated as follows:

$$\dot{Wj} = \frac{r_{ij}^2}{\sum_{j=1}^{m} r_{ij}^2} \tag{3}$$

where $r_{ij}$ is the correlation coefficient of the *j*-th parameter with the baseline parameter Chl-a, and *m* is the number of parameters evaluated.

The correlation coefficients between Chl-a and other parameters for Chinese lake reservoirs are shown in Table 3.

**Table 3.** Correlation coefficients between some parameters and Chl-a in Chinese lakes (reservoirs).

|  | **Chl-a** | **TP** | **TN** | **SD** | **COD$_{Mn}$** |
|---|---|---|---|---|---|
| $r_{ij}$ | 1 | 0.84 | 0.82 | −0.83 | 0.83 |
| $r_{ij}^2$ | 1 | 0.7056 | 0.6724 | 0.6889 | 0.6889 |

Based on the above relationships, the TLI values for each parameter are calculated as follows:

$$TLI(Chl\_a) = 10(2.5 + 1.086 \ln Chl\_a) \tag{4}$$

$$TLI(TP) = 10(9.436 + 1.624 \ln TP) \tag{5}$$

$$TLI(TN) = 10(5.453 + 1.694 \ln TN) \tag{6}$$

$$TLI(SD) = 10(5.118 - 1.94 \ln SD) \tag{7}$$

$$TLI(COD_{Mn}) = 10(0.109 + 2.661 \ln COD_{Mn}) \tag{8}$$

The TLI was calculated using five water quality parameters:

$$\begin{aligned} TLI &= 0.266 TLI(Chl\_a) + 0.1879 TLI(TP) \\ &+ 0.1790 TLI(TN) \qquad\qquad + 0.1834 TLI(SD) + 0.1834 TLI(COD_{Mn}) \end{aligned} \tag{9}$$

## 3. Methodology and Modernization

The contents of this work comprise the following: (1) Using multiple empirical models and in situ measured data to develop regression models about Chl-a and SD, respectively.

(2) Comparing and analyzing the consistency and variability of Chl-a_t obtained from the combination of empirical bands, conc_chl, and in situ measured Chl-a. (3) Comparing the correlation and error size of the results obtained based on the inversion of the better-performing model with experimental Chl-a and conc_chl, Secchi depth, and kd_z90max. (4) Using the in situ measured Chl-a and the retrieval Chl-a to calculate the Trophic level Index (TLI), combining MCI, B5/B4, and B3/B4 to establish a TLI neural network model, and comparing the accuracy parameters (Figure 2).

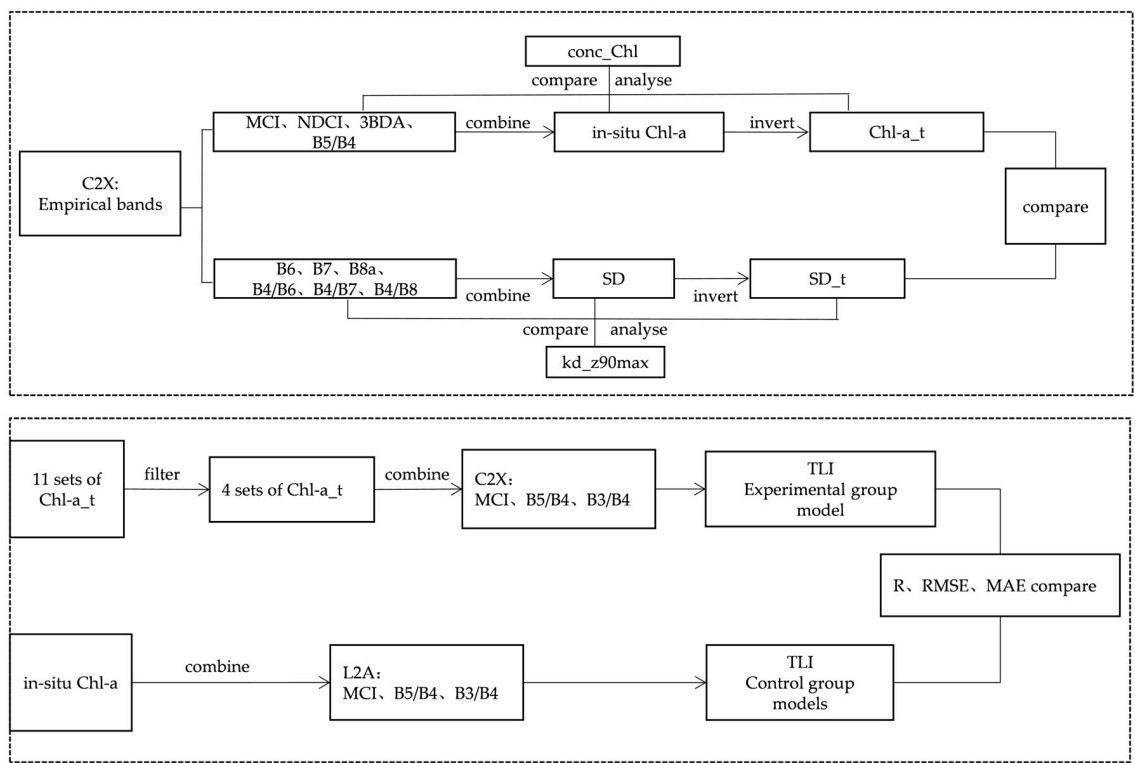

**Figure 2.** Research content framework diagram.

*3.1. Assessment Methods*

Cross-validation is a widely used method in machine learning and statistics for evaluating model performance and tuning parameters. It involves dividing a dataset into subsets or folds, using one part as the test set and the rest as the training set to train and evaluate the model multiple times, resulting in a more reliable performance evaluation. Common cross-validation techniques include k-fold and leave-one-out methods. Due to the limited sample size, a 5-fold cross-validation was employed to assess the model used for water quality estimation in this study. In this study, the training set was partitioned into training and validation samples for iterative modeling and feature column analysis using a nested loop. Within the internal loop, regression models were trained and predicted through cross-validation based on current model types and feature columns.

It is worth noting that although k-fold cross-validation was used in the Chl-a, SD, and TLI modeling above, for the sake of convenience in presenting the fitted regression formulae, what we show was obtained after regressing the entire training set. However, the validation process involved cross-validation to compare the results. The resulting predictions from both training and test samples were then averaged to evaluate subsequent accuracy.

The accuracy indicators used in this study to evaluate the results of all models were correlation coefficient (R), mean absolute error (MAE), and rroot mean squared error (RMSE).

R is a measure of the strength of the linear relationship between the variables and ranges from −1 to 1. The closer it is to 1 or −1, the stronger the linear relationship between the two variables, as well as the better the fit of the model.

MAE reflects the overall level of the model and represents the mean of the absolute errors between estimated and measured values; RMSE illustrates the degree of sample dispersion and represents the sample standard deviation of the difference between predicted and observed values.

$$MAE = \frac{1}{n} \sum_{i=1}^{n} \left| x_{sat_i} - x_{pred_i} \right| \tag{10}$$

$$RMSE = \sqrt{\frac{1}{n} \sum_{i=1}^{n} \left( x_{sat_i} - x_{pred_i} \right)^2} \tag{11}$$

where $x_{sat_i}$ refers to the remotely sensed data obtained in this study as the true value and $x_{pred_i}$ refers to the model-predicted value.

### 3.2. Chl-a Empirical Model

Maximum Chlorophyll Index (MCI) is a crucial indicator for remote sensing of the color of the water that characterizes the Chl-a of cyanobacteria-dominated water bodies. It is the preferred method for remote sensing monitoring of cyanobacteria blooms in inland water bodies [47]. Many researchers [48–50] have utilized the MCI to investigate the inversion of Chl-a. It was found that there is a strong correlation between MCI and Chl-a, allowing effective monitoring of Chl-a in eutrophic lakes. Song et al. [51] obtained good results by using an MCI linear regression approach. McCullough et al. [52] found that MCI is the best predictor of Chl-a.

The Three-band Algorithm (3BDA) has been extensively examined. There is a good relationship between it and Chl-a [53]. Ref. [54] showed that 3BDA could be used for Chl-a concentration estimation from Sentinel-2 data and correlated with Chl-a to 0.94. Xu et al. [55] further showed that the linear regression model of the 3BDA algorithm applies the inversion of Chl-a in inland waters, and Willibroad [13] proved it to be accurate in monitoring Chl-a in Lake Chad.

The Normalized Difference Chlorophyll Index (NDCI) [56] is widely applied to the monitoring of Chl-a. It was shown to be an accurate estimation in Lake Malombe by Rodgers Makwinja et al. [57]. Caballero [37] also demonstrated that NDCI can be used to detect algal blooms and infer Chl-a ranges. NDCI has also been shown to be effective as a Chl-a monitoring tool in a study of two coastal estuaries in the Gulf of Mexico [58].

Additionally, due to the reflection characteristics of water bodies mainly located in the visible light and near-infrared bands, the band combination of B5/B4 is accomplished by associating the ratio of original ground data with Band 5 (red edge) and Band 4 (red) [37]. Chen [59] compared combinations of different band ratios and found that the B5/B4 model is one of the best Chl-a retrieval algorithms. Ge Gao [60]'s Chl-a model, with high accuracy, used a combination of bands such as B5/B4.

In this research, Chl-a models were developed using MCI, NDCI, 3BDA, and B5/B4 as predictors.

$$\text{MCI} = L_W(\lambda_2) - \left[ L_w(\lambda_1) + \frac{\lambda_2 - \lambda_1}{\lambda_3 - \lambda_1} \times (L_w(\lambda_3) - L_w(\lambda_1)) \right] \tag{12}$$

where $L_1$, $L_2$, and $L_3$, respectively, refer to the irradiance with a central wavelength of $\lambda_1$, $\lambda_2$, and $\lambda_3$: $\lambda_1$ = 680.5 nm, $\lambda_2$ = 708 nm, and $\lambda_3$ = 753 nm.

$$\text{NDCI} = \frac{B_1 - B_2}{B_1 + B_2} \tag{13}$$

where $B_1$ and $B_2$ are irradiance levels: $B_1$ = 708 nm and $B_2$ = 665 nm.

$$3BDA = \frac{B_1}{B_2} - \frac{B_1}{B_3} \tag{14}$$

where $B_1$, $B_2$ and $B_3$ are irradiance levels: $B_1$ = 740–750 nm, $B_2$ = 660–670 nm, and $B_3$ = 720–730 nm.

We compared the in situ Chl-a and conc_chl from the entire samples with the empirical band combinations separately to obtain Tables 4–6. It showed that the R (Rc_1) values between 3BDA, MCI, NDCI, B5/B4, and the in situ Chl-a were higher than 0.2, surpassing those of other band combinations. This finding was consistent with previous studies and confirmed their efficacy. Moreover, the correlation trends between 4 band combinations with conc_chl were similar to those observed between measured Chl-a levels; notably, higher correlations were found for 3BDA, NDCI, and B5/B4, while the lowest correlation was observed for MCI. The R (Rc_2) between 3BDA, MCI, NDCI, B5/B4, and conc_chl (Rc_2) were higher than 0.65. The difference (Rc_dif_1) between Rc_1 and Rc_2 had values in the range of 0.45–0.65.

**Table 4.** R of band combinations with in situ Chl-a and conc_chl.

| R | 3BDA | MCI | NDCI | B5/B4 |
|---|---|---|---|---|
| In situ Chl-a | 0.32 | 0.23 | 0.36 | 0.34 |
| conc_chl | 0.98 | 0.66 | 0.92 | 0.96 |

**Table 5.** RMSE of band combinations with in situ Chl-a and conc_chl.

| RMSE (µg/L) | 3BDA | MCI | NDCI | B5/B4 |
|---|---|---|---|---|
| In situ Chl-a | 37.19 | 37.29 | 37.19 | 36.24 |
| conc_chl | 30.13 | 30.22 | 30.13 | 28.99 |

**Table 6.** MAE of band combinations with in situ Chl-a and conc_chl.

| MAE (µg/L) | 3BDA | MCI | NDCI | B5/B4 |
|---|---|---|---|---|
| In situ Chl-a | 32.43 | 32.57 | 32.44 | 31.440 |
| conc_chl | 27.47 | 27.56 | 27.47 | 26.33 |

The dependent variable was in situ Chl-a, while 3BDA, MCI, NDCI, and B5/B4 served as independent variables. Different mathematical functional forms, including linear (LIN), logarithmic (LOG), exponential (EXP), and power (POW) functions, were examined to determine the optimal derived model by considering the effect of positive and negative amplitudes for different combinations of bands. Based on the magnitude of 3BDA, MCI, NDCI, and B5/B4 values, we derived 11 regression equations (Table 7) to fit the Chl-a model inversion results. Henceforth, Chl-a_t would be used as a descriptor for these results.

**Table 7.** In situ Ch-a and bands to create regression equations.

| | Linear | Exponential | Logarithmic | Power |
|---|---|---|---|---|
| 3BDA | y = 49.25x + 23.654 | | | |
| MCI | y = 1785.5x + 19.914 | $y = 12.965 \times 10^{94.593x}$ | y = 7.6841ln(x) + 70.814 | $y = 208.07x^{0.4214}$ |
| NDCI | y = 69.621x + 21.985 | $y = 14.867 \times 10^{3.3951x}$ | | |
| B5/B4 | y = 24.973x − 2.3716 | $y = 4.4626 \times 10^{1.2305x}$ | y = 33.858ln(x) + 22.068 | $y = 14.922x^{1.6531}$ |

### 3.3. SD-Based Empirical Modeling Aids Validation

Through analysis of C2RCC products, kd_z90max has the best correlation with SD, which has been successfully used in the retrieval and verification of water-quality products [43]. The kd_z90max product can be used to estimate the SD of optically complex

waters and can provide effective information for SD research. Previous studies have shown that the in situ Chl-a and conc_chl have an RMSE of 3.73 µg/L and a Normalized Root Mean Squared Error (NRMSE) of 19%, while the in situ SD and kd_z90max have an RMSE of 2.26 m and an NRMSE of 14%. It has been preliminarily confirmed that the robustness between kd_z90max and SD was stronger.

Compared with Chl-a, SD has obvious advantages: (1) it can be identified and obtained by the naked eye in real-time without sending water samples to the laboratory; (2) the measurement method is relatively simple and has no experimental error, which might involve less experimental error; and (3) it can be measured in a wide range, with high synchronization with image passing time. The measurement of Chl-a involves a series of operations, including sample pre-processing and experimental analysis, which are easily interfered with by external factors, resulting in more errors. In addition, the SD values in water bodies exhibit greater stability [61–64]. Physical coefficients are known to change more dramatically in the water column over short periods of time and can cause corresponding changes in water quality parameters, and it has been shown that the correlation between changes in Chl-a caused directly by physical coefficients is greater than that caused by changes in SD [65]. Changes in Chl-a are mainly influenced by algae and phytoplankton, and SD is mainly influenced by suspended matter, while daily changes in the former have a larger variation factor than the latter [66]. Thus, although Chl-a changes dramatically over a short time, SD does not change synchronously [67,68]. From this perspective, the acquisition time for SD is better synchronized with image transit time. Overall, SD can provide a relatively reliable reference value.

Therefore, in situ SD could theoretically be used as a water quality parameter with a much smaller data error to validate the data error of in situ Chl-a. To leverage the higher accuracy of SD, we developed empirical models of SD to assist validation and assess the uncertainty of in situ Chl-a. By building the model by the same empirical regression and comparing the difference between the Chl model and the SD model, the error of Chl can be judged based on the results of the SD model.

Research data show that except for the water vapor band (B9) and the short-wave infrared band (B11, B12), the first nine bands have high correlation coefficients with transparency [69–71]. The R analysis between SD (in situ data and kd_z90max) and the empirical band combinations were conducted, respectively. Table 8 showed that the correlation between band combinations and in situ SD was also consistent with previous studies.

**Table 8.** R between band combinations and in situ SD.

| R | B6 | B7 | B8a | B4/B6 | B4/B7 | B4/B8 |
|---|---|---|---|---|---|---|
| In situ SD | −0.35 | −0.35 | −0.36 | 0.42 | 0.42 | 0.41 |
| kd_z90max | −0.72 | −0.73 | −0.74 | 0.91 | 0.92 | 0.92 |

The R (Rd_1) between single bands 6, 7, 8A, B4/B6, B4/B6, B4/B8, and in situ SD data were all higher, respectively, far exceeding the R (Rc_1) of the empirical band combinations with the in situ measured Chl-a. The R (Rd_2) between bands and kd_z90max data were higher than 0.7. The absolute values of the correlation between each band combination and kd_z90max were higher than the absolute values of the R with SD. The difference (Rd_dif_1) between Rd_1 and Rd_2 ranged from 0.4 to 0.5, which was better than the Chl-a case, where Rd_dif_1 was about 0.1 lower than the Rc_dif_1.

The regression analysis for SD was conducted by combining six band combinations (bands 6, 7, 8A, B4/B6, B4/B6, and B4/B8) based on the training sample. Similarly, in situ SD was used as the dependent variable, and band data as the independent variable. With linear, exponential, logarithmic, and power functions, 24 equations (Table 9) were finally obtained by each regression fit between remote sensing information and SD. Each combination of bands for the test sample group was used as the dependent variable to obtain the SD inversions (SD_t).

**Table 9.** In situ SD and bands to create regression equations.

|  | Linear | Exponential | Logarithmic | Power |
|---|---|---|---|---|
| B6 | $y = -5552.5x + 77.625$ | $y = 70.359 \times 10^{-88.57x}$ | $y = -23.27\ln(x) - 76.599$ | $y = 8.0485x^{-0.319}$ |
| B7 | $y = -5385.7x + 76.981$ | $y = 69.863 \times 10^{-86.65x}$ | $y = -23.07\ln(x) - 75.59$ | $y = 8.1212x^{-0.317}$ |
| B8a | $y = -12646x + 75.332$ | $y = 68.363 \times 10^{-206.3x}$ | $y = -23.06\ln(x) - 97.048$ | $y = 5.9927x^{-0.318}$ |
| B4/B6 | $y = 19.105x + 8.2487$ | $y = 25.616 \times 10^{0.2646x}$ | $y = 42.345\ln(x) + 18.999$ | $y = 28.968x^{0.6177}$ |
| B4/B7 | $y = 18.135x + 10.176$ | $y = 26.355 \times 10^{0.2504x}$ | $y = 39.9\ln(x) + 20.88$ | $y = 29.748x^{0.5831}$ |
| B4/B8 | $y = 6.4456x + 14.062$ | $y = 27.73 \times 10^{0.0895x}$ | $y = 35.36\ln(x) - 8.3866$ | $y = 19.284x^{0.52}$ |

*3.4. Assisted Verification-TLI-Based Modeling*

TLI is an assessment method that takes multiple Water Quality Parameters (WQPs) into account, resulting in more compelling outcomes for water quality evaluation. Researchers have employed TLI to assess the intensity of eutrophication, the health status of aquatic ecosystems, and water quality classification, demonstrating its ability to provide more comprehensive and reliable results for assessing water quality [72–74].

Although some factors affecting water quality have a good linear relationship with the water spectral characteristics, there are still WQPs that cannot be simply described by linear or nonlinear methods. The neural network of machine learning can describe nonlinear and complex relationships between independent variables and dependent variables, which have been widely used in water quality estimation research. The Radial Basis Function Network (RBFNN) has the features of simple structure, strong nonlinear approximation ability, fast convergence speed, strong generality, etc. These features make it an effective tool for enhancing the accuracy of water quality monitoring [75–77]. A TLI inversion model was established using RBFNN to capture the nonlinear relationship between WQP and remote sensing information. The inputs of the model were MCI, B5/B4, and B3/B4, while TLI obtained from WQP field water quality monitoring data served as the output. This approach has been proven effective in inverting TLI and providing important support for water quality monitoring and management [33,78]. Therefore, we adopted the same methodology to model TLI in this study.

The Chl-a_t was derived by comparing the 11 groups of regression fits in Section 3.2 with the in situ Chl-a and conc_chl, and selecting the four groups of Chl-a_t that exhibited superior correlations (R, RMSE, and MAE) for each band combination of MCI, NDCI, 3BDA, and B5/B4. The screened four groups of Chl-a_t were then combined with SD, TN, TP, and $COD_{Mn}$ from the sample library to create an experimental sample pool using MCI, B5/B4, and C2X-corrected B3/B4 calculations. The MCI, B5/B4, and B3/B4 calculated from the L2A level image data, in conjunction with the TLI derived from in situ measured Chl-a, were utilized as a reference sample pool. To model the TLI, we employed a cross-validation approach and compared training and test set outcomes.

The *newrb* function in MATLAB 2018b was used to train the network. The main steps were as follows:

(1) The MAPMINMAX function was applied to normalize input and output data to $-1$ and 1;

(2) Objectives and network training parameters were set. Data from the training set were used to train the network. The number of neurons in the hidden layer was iterated from 0 to 30;

(3) The data from the test set were then used to test the network. The number of neurons was adjusted to optimize the network. Compare the results from different numbers of neurons.

Note: A rule that the number of neurons should not exceed 30 was set to avoid over-fitting.

## 4. Results

### 4.1. Analysis of Chl-a Results

By comparing the in situ measured Chl-a with the conc_chl in all samples, we observed a strong correlation in areas of lower concentration, but a weak correlation in regions of concentration areas and an overall poor correlation. As the conc_chl has been validated and applied by many parties, the poor R between the in situ Chl-a and its correlation was a preliminary indication of the existence of data error (Figure 3).

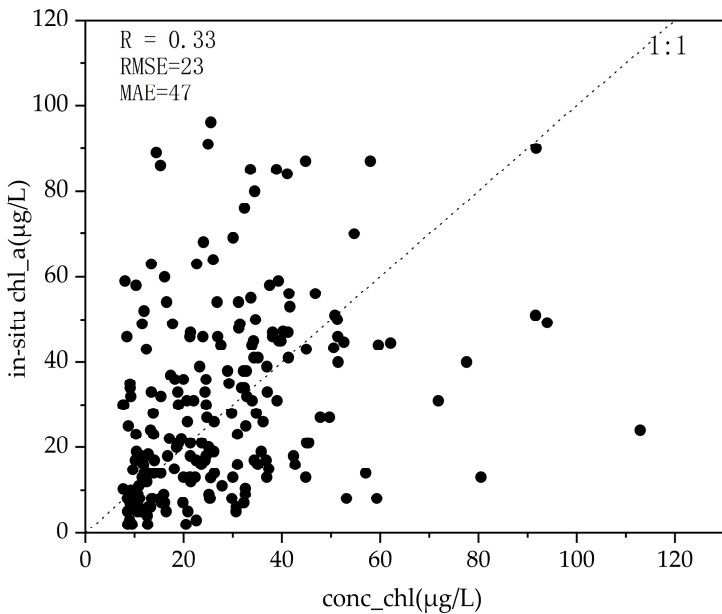

**Figure 3.** Comparison of in situ Chl-a, conc_chl.

The concentration range was divided based on conc_chl. Since in situ Chl-a was not detected below 2 μg/L and above 10 μg/L in most areas of Wuhan, the boundaries of 2 μg/L and 10 μg/L were used. The comparison range is then divided according to the sample size, using 20, 30, 40, 50, and 120 μg/L as bounds to compare conc_chl with in situ measured Chl-a better. As shown in Table 10, as the range of conc_chl increased, RMSE and MAE did not necessarily become smaller with it. The smallest values of RMSE and MAE were found in the conc_chl range between 2 and 10 μg/L, indicating the best correlation and smallest error between the conc_chl and the corrected Chl-a_t this time. Higher values of RMSE and MAE were found in the conc_chl range between 50 and 120 μg/L when the conc_chl correlated poorly with in situ measured Chl-a and had higher errors.

**Table 10.** Comparison of different ranges of in situ Chl-a with conc_chl.

| Range μg/L | No. of Samples | R | RMSE (μg/L) | MAE (μg/L) |
|:---:|:---:|:---:|:---:|:---:|
| 2~10 | 20 | −0.40 | 19.08 | 13.29 |
| 10~20 | 62 | 0.16 | 21.04 | 13.27 |
| 20~30 | 46 | 0.16 | 21.76 | 15.83 |
| 30~40 | 46 | 0.20 | 20.69 | 16.49 |
| 40~50 | 16 | −0.30 | 23.15 | 19.38 |
| 50~120 | 20 | 0.02 | 37.16 | 34.89 |

In the Chl group, the accuracy parameters of R, RMSE, and MAE were compared between Chl-a_t obtained from the fitted equation and other data (conc_chl and in situ Chl-a) involved in the calculation. The regression models were evaluated by comparing these accuracy parameters' magnitudes and establishing a relationship between conc_chl and in situ Chl-a in both training and test sample groups. Regarding Chl-a modeling, we presented

the average accuracy parameters obtained through cross-validation to demonstrate the rigor of our experiment. Furthermore, by comparing the results of training and test samples, we can evaluate the generalization ability of our fitting approach. Therefore, we compare the training and test sets based on their original division during modeling.

Comparing conc_chl and in situ Chl-a, the majority of the fitting methods were relatively close on the training and test samples. Additionally, the performance of both the training and test sets shows that the predictions of the four band combinations are closer to the conc_chl data, mainly in that the Chl-a_t correlation with the conc_chl product (Rc_3) was in the range of 0.5–0.9, the MAE was around 10–15 μg/L, and the RMSE was around 8–10 μg/L, while for the correlation with the in situ Chl-a correlation (Rc_4), the RMSE was around 15 μg/L and the MAE is around 20 μg/L.

From an intuitive perspective, it was evident that there exists a significant difference (Rc_dif_2) between the values of Rc_3 and Rc_4, with the range of Rc_dif_2 being approximately 0.4 to 0.7 within the training sample group. In the test sample group, Rc_dif_2 ranged from 0.2 to 0.4. Compared to the relationship between in situ Chl-a and the empirical band combinations, in situ Chl-a presented a stronger correlation with the Chl-a_t. In Figure 4, the fitted prediction results for Chl-a_t data exhibited lower RMSE and MAE values compared to in situ Chl-a, with both RMSE and MAE decreasing. Additionally, conc_chl was more closely correlated with Chl-a_t, which also displayed a decrease in RMSE and MAE values relative to empirical band combinations. The RMSE of Chl-a_t using the conc_chl product was more than halved compared to that using in situ Chl-a. Similarly, the MAE of Chl-a_t and conc_chl products was more than halved compared to in situ Chl-a.

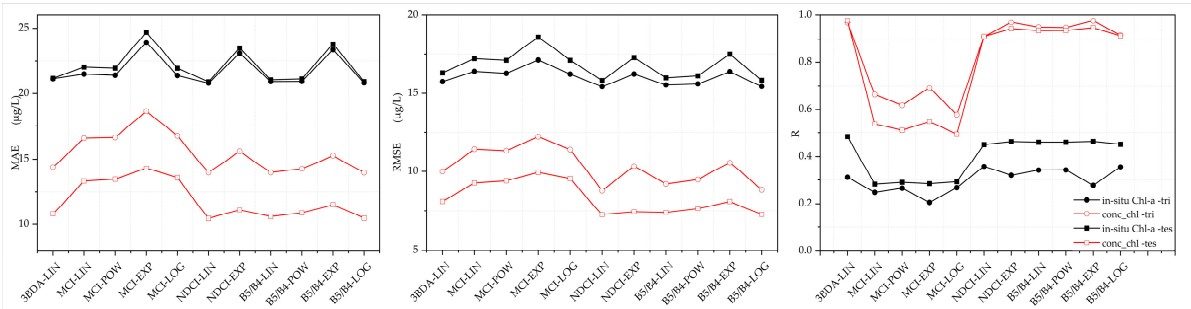

**Figure 4.** Accuracy parameters of the Chl-a model.

Overall, for the correlation between the in situ Chl-a and the Chl-a_t results, the RMSE and MAE have all changed from the original combinations of the bands.

For the purpose of TLI modeling, it was necessary to compare the results of the different fits within each band combination, and it can be observed from the given data that the different fits perform slightly differently on these metrics. Specifically, NDCI-LIN exhibited higher correlation coefficients and lower RMSE and MAE in the test samples, indicating that it was able to predict changes in concentration chlorophyll more accurately. Similarly, we ultimately chose to find the Chl-a_t results for the 3BDA linear fit, the MCI power function fit, the NDCI linear fit, and the B5/B4 logit fit for the TLI, respectively.

### 4.2. Analysis of SD Results

Based on the above argument (Section 3.3), it can be concluded that the in situ SD and kd_z90max are validated and highly accurate measurement indexes, which are used to measure the turbidity of water. Empirical band combinations are models that have been scientifically studied and empirically analyzed to estimate Chl-a and SD. By comparing R between data, we can mine relationships between data. When we compared the in situ SD and kd_z90max with the empirical band combination, we found that there is little correlation difference between them, which means that they have similar accuracy in measuring SD and are closely related. Similarly, based on the correlation between SD model

data, the correlation gap between the Chl-a model and SD model data can be compared to find out whether the Chl-a data relationship is close. Therefore, it is important to pay attention to R in the SD model.

Of all the band combinations, the R between the two band combinations B4/B6 and B4/B7 was higher than the others. The R between the B4/B6 linear fit and kd_z90max was 0.93, while the R between the other band combinations was generally below 0.9 but still higher than 0.7 (Figure 5).

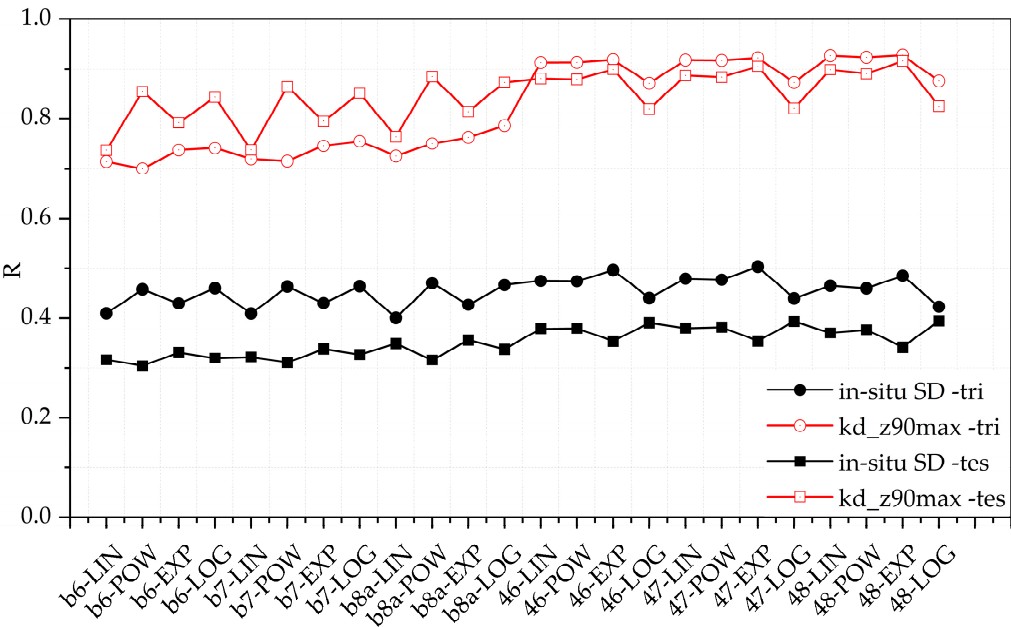

**Figure 5.** Comparison of SD_t with in situ SD and kd_z90max correlation.

The in situ SD correlation with SD_t was designated as Rd_3, the kd_z90max correlation with SD_t was designated as Rd_4, and the discrepancy between Rd_3 and Rd_4 was denoted as Rd_dif_2.

The variability of the training and test sets remains consistent, indicating that the SD_t results obtained from the regression analysis equation are more strongly correlated with in situ SD than band combinations are with in situ Chl-a. Figure 5 shows that Rd_3 ranged from 0.3 to 0.5, Rd_4 ranged from 0.7 to 0.9, and Rd_dif_2 has a value in.0.35–0.5. The Rd_4 was higher than Rd_3, and these two correlations are higher than the Rc_3 and Rc_4.

Rd_dif_1 is lower than the value of Rc_diff_1; also, Rd_dif_2 was lower than Rc_dif_2.

### 4.3. Analysis of TLI Modeling Results

There exists a direct relationship between the TLI and Chl-a, SD, TN, TP, and $COD_{Mn}$. While the TLI can reflect the changes in these parameters, observing the trend of TLI changes can determine whether there are errors in water quality parameters. Accordingly, when involved in the TLI calculation of individual water quality parameters, measurement errors can affect the overall accuracy and stability of TLI. Fixing all other water quality parameters and adjusting only the Chl-a values allows for the modeling of the TLI. By observing changes in the model accuracy parameters, errors in the in situ Chl-a could be assessed.

The TLI calculated from these four sets (3BDA linear fit, the MCI power function fit, the NDCI liner fit, and the B5/B4 logarithmic fit) of Chl-a_t values were chosen as the output parameters (Figure 6).

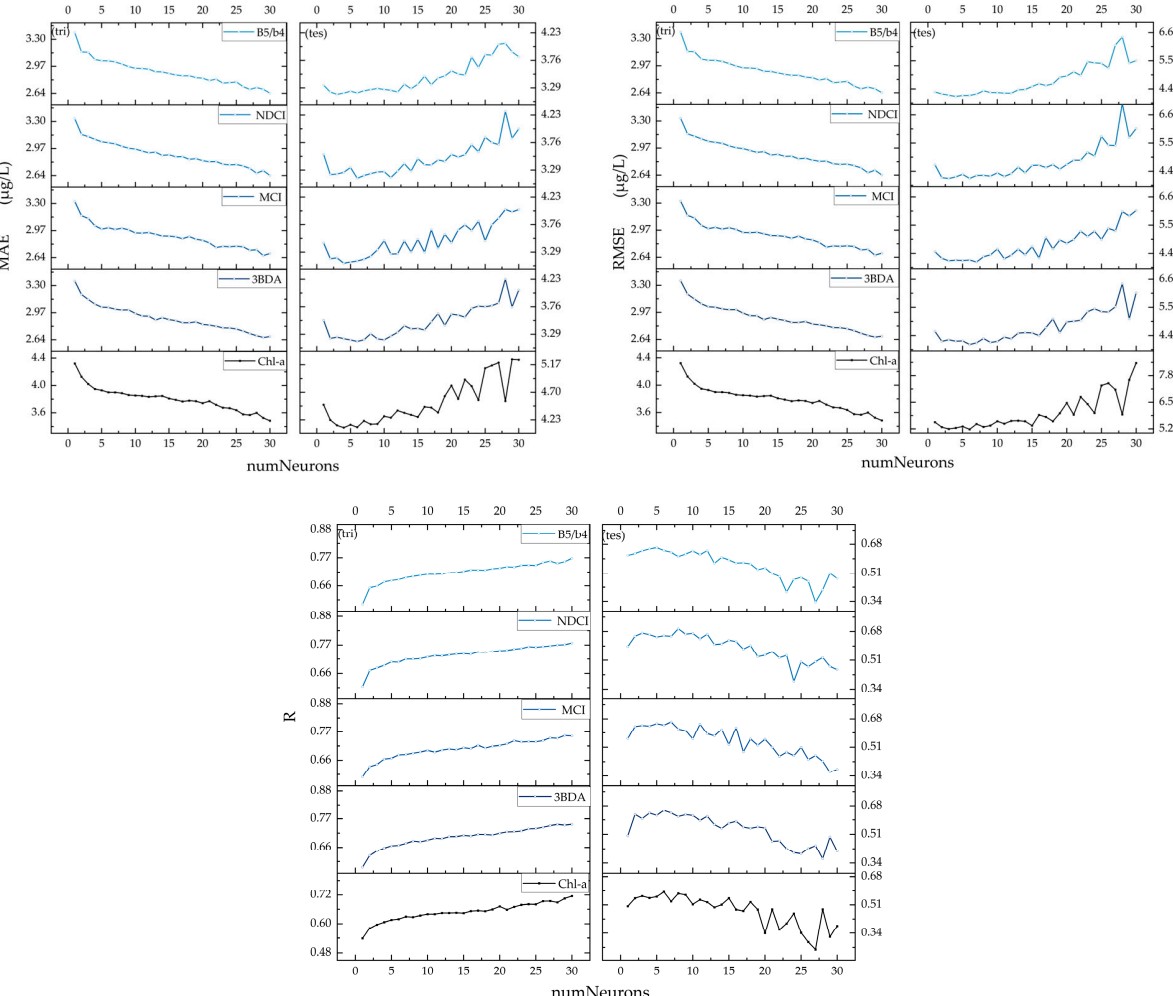

**Figure 6.** Accuracy parameters for TLI model results.

Distinguishing the test set from the training set, the training set performed more smoothly, and the test set had a smaller sample size and a larger band amplitude. Overall, the consistency in the number of neurons ranging from 1 to 30 implied that when comparing the TLI model constructed by in situ Chl-a, there was a stronger correlation between the TLI calculated by the Chl-a_t in the experimental sample pool and that predicted by the model, than between actual the Chl-a and their corresponding predicted TLI in the control set. The accuracy of the chlorophyll prediction model and the error of the TLI parameter obtained by four empirical belt fitting methods were all smaller than that of the original TLI model. The MAE, RMSE, and R of the original TLI were 4.47, 5.84, and 0.48, respectively. In the fitted TLI model, the average MAE, RMSE, and R of each group were 3.45, 4.65, and 0.55, respectively. The calculated TLI of Chl-a_t of each group was about one time smaller than the original TLI, and the R was also larger. In the same test set, the number of different neurons was compared. The maximum MAE of the original TLI was 5.20, the minimum MAE of the TLI was 4.1, the maximum RMSE of the TLI was 7.4, and the minimum RMSE of the TLI was 5.2. The maximum MAE of the four fitted TLI groups was about 4.1, and the minimum was 3.1. The highest RMSE was 6.5, and the lowest was 4.1.

Whereas the different number of neurons made the results slightly change in different modeling combinations, analysis of the fluctuations in the folds also showed that the TLI modeling accuracy fluctuates the most with the involvement of in situ Chl-a.

## 5. Discussions

From the results, we can see the errors in in situ Chl-a model construction. Reducing measurement errors is crucial for water body monitoring and management, as accurate Chl-a data allow for a better understanding of spatial distribution and dynamics. It supports the establishment of prediction and spatio-temporal simulation models for Chl-a trends in water bodies, providing reliable data for water resources management, scientific research, environmental protection, and decision making.

In this study, the image data were downloaded and processed to minimize timing errors caused by objective reasons, and we filtered the image dates to keep the transit times within 3 days of the sampling time. The empirical bands used in the analysis have been validated by many authors with large amounts of data, and the conc_chl product has been trained to be more reliable with a large amount of data, ensuring consistent results for Chl-a within the range required by SNAP 9.0. Additionally, SNAP's water quality parameter model utilizes neural networks to capture complex data patterns and interactions through nonlinear relationships. Neural networks play a crucial role in water quality parameter calculations. By adopting the neural network approach, C2X can effectively train models with large datasets to automatically adjust weights and parameters based on complex features and correlations, thereby extracting key feature information for the accurate prediction of water quality parameters. Compared with the influence of whether the waveband involved in product generation overlaps with the waveband used in this study, the expression of the linear relationship is weak. This means that the neural network can better deal with the nonlinear relationship, whether there are overlapping bands or not, and extract effective features and patterns from the data to then calculate water quality parameters.

However, there are still some limitations and areas for improvement in this study, and the uncertainty of in situ Chl-a has not yet been completely resolved. The value of the conc_chl product is small, but it can be refined in future studies by adjusting the coefficients in the formula to bring it closer to actual values. The large values of RMSE and MAE obtained for the corrected Chl-a_t and band combinations reflect the low stability of the Chl-a_t data, which may be attributed to poor image quality. In locations where clear areas cannot be obtained due to poor image quality, errors increase accordingly. Future research will focus on improving the accuracy of Chl-a_t data and collecting more comprehensive chlorophyll concentration data from ground-based sample sites with the help of official SNAP published information. At the same time, it is expected to rely on more advanced and high-precision instrumentation to improve the precision and accuracy of Chl-a measurements and to be able to make reliable measurements at lower Chl-a. In terms of model construction, the establishment of accurate calibration models that consider multiple influencing factors such as environmental factors, light levels, and water quality parameters, combined with machine learning and artificial intelligence techniques to develop more accurate data processing algorithms that can identify and correct for noise and interference in measurements can improve the accuracy of Chl-a measurements. In addition, the establishment of an internationally harmonized and standardized methodology will facilitate result comparison and data sharing among different laboratories. This will further enhance data processing and modeling methods to improve the accuracy of Chl-a measurements, promoting the development of water body monitoring and management.

## 6. Conclusions

Overall, the modeling results indicate that the change trends in both the training and test sets based on the given data are similar, suggesting good generalization to predicted data. Additionally, cross-validation was employed to mitigate any potential bias from artificially selected data, thus ensuring the reliability of our study. The detailed conclusions are as follows:

With few exceptions, the conc_chl product consistently reflected the measured Chl_a concentrations across the sample pool. Compared to in situ Chl-a, conc-chl exhibited a

stronger correlation and less error across fitting methods. The relationship between conc-chl and these methods was found to be more robust, indicating its reliability as a Chl-a indicator. In comparison with conc_chl, in situ Chl-a had a lower error parameter associated with empirical bands, suggesting that in situ data were less reliable than conc_chl.

Based on the extensive certification of the empirical band by numerous scholars, as well as the widespread application and training of conc_chl and kd_z90max to a vast array of datasets, their relationship with real-world data has been thoroughly confirmed and accurately modeled. Therefore, in situ data that closely align with these parameters can be considered highly authentic representations of actual products. Conc_chl and kd_z90max have been trained and applied by numerous data applications, and the relationship with real data has been widely confirmed and accurately modeled. Additionally, the remote sensing reflectance (Rc_1) of in situ spectral data using the empirical band combination was higher than that of Chl_a estimated with the same band combination. The correlation gap (Rc_diff_1) between the band combination and conc_chl product as well as in situ Chl_a was larger than Rd_diff_1 between the band combination and kd_z90max and in situ spectral data. This illustrated the poor accuracy of in situ Chl-a compared to in situ SD.

Further comparison of the SD modeling results with the Chl-a modeling results reveals that, after correcting for empirical regression, Rd_3 exhibited a smaller difference from baseline data than Rc_3 did. Additionally, Rc_diff_2 showed a larger discrepancy compared to Rd_diff_2. Therefore, it can be inferred that in situ Chl-a measurements are subject to greater error than in situ SD measurements. The correction process effectively reduces errors in SD data and brings them closer to actual conditions. The significant discrepancy between the Chl-a correction results and the baseline data suggests that Chl-a measurements were influenced by multifaceted factors, and the correction process had not entirely rectified errors in Chl-a. Therefore, it is crucial to acknowledge inaccuracies in measured Chl-a data when utilizing or interpreting such information.

Further Chl-a modeling analysis revealed that the predictions of Chl-a_t, when corrected for in situ using C2X atmospheric corrected band data, exhibited improved accuracy compared to empirical band combinations. Specifically, the model demonstrated a substantial decrease in both RMSE and MAE values, indicating a reduction in error between predicted results and actual observations. The TLI modeling results were also analyzed, and it was found that the band-corrected Chl_a concentrations were involved in the TLI calculation, resulting in a decrease in both RMSE and MAE of the TLI model compared to the original TLI model calculated by in situ Chl-a. This suggests that the stability of the TLI model was also improved by the correction results. Additionally, the R-value of the model increased, and a higher correlation coefficient indicated a stronger linear relationship between predicted values and actual observations, thus increasing the reliability of the model. The above model results all illustrate that empirical band correction can effectively reduce this error. It also allows the correction of the C2X product in the model inversion to be confirmed.

This study overall confirms that in situ Chl-a data are prone to errors and cannot be fully relied upon, but the modeling correction based on the C2X band significantly enhances the reliability of Chl-a data.

**Author Contributions:** Conceptualization, W.L. and B.H.; methodology, W.L. and B.H.; software, W.L.; validation, W.L.; formal analysis, W.L. and B.H.; investigation, F.Y. and C.F.; resources, F.Y., C.F. and X.Y.; data curation, W.L., F.Y. and C.F.; writing—original draft preparation, W.L.; writing—review and editing, B.H., Y.Z. and H.L.; visualization, W.L.; supervision, B.H., Y.Z. and H.L.; project administration, B.H.; funding acquisition, B.H. All authors have read and agreed to the published version of the manuscript.

**Funding:** This work was funded by the Key scientific research projects of the Hubei Provincial Department of water resources (grant number: HBSLKY202127); Chinese Academy of Sciences Strategic Pioneer Science and Technology Special Project (Class A) Beautiful China Ecological Civilization Science and Technology Project (grant number: XDA23040201); Hubei Provincial Key R&D Program Project (grant number: 2020BCB074); Hubei Provincial Natural Science Foundation of China (grant number: 2022CFB857); and Knowledge Innovation Program of Wuhan-Basic Research (grant number:2022010801010135).

**Informed Consent Statement:** Informed consent was obtained from all subjects involved in the study.

**Data Availability Statement:** The situ data are not publicly available due to privacy. The Sentinel-2 data used in this study were downloaded from the Sentinel Scientific Data Hub (https://scihub.copernicus.eu/ accessed on 1 January 2022).

**Acknowledgments:** The authors would like to acknowledge the support provided by the above fund programs. The authors are grateful to Qi Feng for his help in data collection.

**Conflicts of Interest:** The authors declare no conflict of interest.

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
