# Peer review of "Using C2X to Explore the Uncertainty of In Situ Chlorophyll-a and Improve the Accuracy of Inversion Models"

_sustainability, doi:10.3390/su15129516_

Round 1
Reviewer 1 Report
This paper evaluates the uncertainties on chlorophyll a content in a lake in Wuhan, China using multiple methods. The authors tested two different atmospheric correction methods, empirical models, and in situ measurements. The experiments sound interesting. However, I do not understand the scientific hypothesis that the authors are trying to test. The organization of the paper is not very clear. Given the major comments below, I do not think the manuscript is ready for publish.
Major comments:
-
None of the acronyms used in this paper are explained when they first appear.
-
Please clarify what is the knowledge gap you are trying to solve. For example, atmospheric correction, empirical model, or uncertainties from fast changes caused by weather and environment between satellite observations?
-
The in situ sample size is very small. How many samples are left after the outliers are taken out? Is there any cross-validation being used?
-
Figures with better quality and resolutions are needed.
Minor comments:
-
L32. what do you mean by “transiting” ?
-
L50. what is referred by “it” and “its”?
-
L63 what do you mean by “nervous”?
-
2.3.3. What do different TLI’s represent physically?
Some proof reading is needed.
Reviewer 2 Report
sustainability-2400316-peer-review-v1
Article: Using C2X to explore the uncertainty of in-situ chlorophyll-a and improve the accuracy of inversion models.
The idea of the manuscript is unique, new, and complete for publication, with minor corrections
Abstracts:
Line 18: RMSE and MAE correct to Root Mean Squared Error (RMSE) and Mean Absolute Error (MAE).
Introduction:
The introduction was written in a unique way.
Materials and Methods:
Research materials and methods were written in a clear and well-written manner
Results:
The results are presented in a distinctive way, but the Figures need improvement
Discussion: Presented appropriately with the manuscript, but you need support with different sources.
Conclusions: good section.
Reviewer 3 Report
Overall, this paper provides a detailed analysis of the use of remote sensing data to monitor water quality in small and medium-sized water bodies, with a specific focus on the lakes in Wuhan, China. The authors have done a thorough job of describing the study area, the methods used to acquire and process remote sensing data, and the collection of ground sample points to obtain water quality component data. The results of their analysis are presented clearly and the conclusions drawn are well-supported by the data.
However, there are a few areas where the paper could be improved. Firstly, the introduction could be expanded to provide more background on the importance of monitoring water quality in small and medium-sized water bodies, and why the study area of Wuhan was chosen. Secondly, the language in some sections of the paper could be improved to make it more concise and easier to follow. Finally, the paper would benefit from a more detailed discussion of the implications of the findings and potential avenues for future research.
Abstract:
1. Provide more background information at the beginning of the abstract, such as the role and importance of Chl-a.
2. Describe the C2X and C2RCC methods in more detail to help readers better understand the study design.
3. Provide more details and numbers in the results section to help readers better understand the data and analysis.
4. The results section mentions two correlation coefficients R1 and R2 for Chl-a but does not explain the meaning and usage of these coefficients. The author needs to provide more information to help readers understand these results.
5. Some terms may not be familiar to general readers, such as "conc_chl" and "TLI". It is necessary to explain the meaning of these terms to help readers better understand the article.
6. The abstract does not mention specific research conclusions, only some results. The author needs to summarize and explain the meaning of these results in the conclusion section.
Introduction:
One suggestion for improvement could be to provide more specific information on the research question or objective of the study earlier in the introduction. While the final paragraph mentions the aim of the study to investigate the uncertainty associated with in-situ Chl-a and improve its accuracy, this information could be included earlier in the introduction to give the reader a clearer understanding of the focus of the study. Additionally, it may be helpful to briefly explain what is meant by "empirical regression equations" and "TLI neural network modeling" to provide more context for the methods used in the study.
Results:
I would suggest increasing the resolution of all figures in the manuscript, as the current version is too blurry.
Discussions:
The manuscript would benefit from a more detailed discussion of the implications of the findings and potential avenues for future research.
Conclusion:
The conclusion section is important to leave a lasting impression. Make sure that your conclusion is clear, concise, and well-written.
Round 2
Reviewer 1 Report
The revision is nicely done. I do not have further comments. Congrats.
Reviewer 3 Report
This version of the manuscript has been greatly improved. I have no more comments.